

# A longitudinal study of risk factors associated with white spot disease occurrence in marine shrimp farming in Rayong, Thailand

Sompit Yaemkasem[1,2,3], Visanu Boonyawiwat[4], Manakorn Sukmak[4], Sukanya Thongratsakul[2] and Chaithep Poolkhet[2]

[1] Graduate Program of Animal Health and Biomedical Science, Faculty of Veterinary Medicine, Kasetsart University, Bangkok, Thailand
[2] Section of Epidemiology, Faculty of Veterinary Medicine, Kasetsart University, Kamphaeng Saen, Nakhon Pathom, Thailand
[3] Department of Fisheries, Ministry of Agriculture and Cooperatives, Rayong Coastal Aquaculture Research and Development Center, Rayong, Thailand
[4] Section of Aquatic Animal Diseases, Faculty of Veterinary Medicine, Kasetsart University, Kamphaeng Saen, Nakhon Pathom, Thailand

Corresponding author
Chaithep Poolkhet, fvetctp@ku.ac.th

## ABSTRACT

**Background:** A longitudinal study was conducted to analyze farm characteristics, farm practices, and biosecurity measures that influenced the occurrence of white spot disease (WSD) in shrimp farming in Rayong, Thailand.

**Methods:** Data were collected using a structured interview schedule administered between October 2017 and September 2019. A generalized estimating equations (GEE) model was used to identify risk factors. From the 270 responses, 86 possible risk factors were analyzed using univariate and multivariate analysis.

**Results:** We found that 17 possible risk factors were statistically significant with an alpha level of 1% and associated with WSD status. In the final model, multivariate analysis found that two independent variables were statistically significant. First, the absence of inclement weather during ponding was a statistically significant factor associated with WSD occurrence and it prevented WSD occurrence as the odds ratio (OR) was <1 (OR = 0.196, $P = 1.3 \times 10^{-5}$). Second, the separation of ponds into three specific types (cultured, water treatment, and pond for water reservoir) was significantly associated with WSD occurrence. Likewise, the presence of this ponding system was a protective factor against WSD (OR = 0.0828, $P = 0.001$).

**Conclusions:** The results of this study offer a reference for farmers and relevant authorities when addressing WSD occurrence in shrimp farming. In addition, our results can help relevant authorities in controlling WSD in other endemic areas.

# INTRODUCTION

White spot disease (WSD), a severe viral disease affecting marine shrimp farming, is caused by the white spot syndrome virus (WSSV). This double-stranded DNA virus belongs to the genus Whispovirus within the Nimaviridae family. WSSV has a wide host

range, including the marine shrimp, and it can be transmitted vertically and horizontally. The mortality rate of infected penaeid shrimp is high; however, the morbidity rate varies depending on various factors (*OIE, 2009*). In general, the cumulative mortality rate can reach 90–100% within 3–10 days after infection (*Wang et al., 1999*; *Millard et al., 2021*). In infected shrimp with WSD, lesions that appear as white spots are commonly found within the exoskeleton. Furthermore, spots are present on the carapace of shrimp; while they are normally white, the colour of these spots can vary. Reddish or pinkish discoloration of shrimp is always present (*OIE, 2019*). Infected shrimp usually present abnormal behavior such as lethargy, slow swimming, swimming on the water surface, swimming along the edge of the pond, and/or reduced feed intake (*OIE, 2019*). Infrequent, infected penaeid shrimp present subclinical disease. Therefore, laboratory confirmation is needed to confirm WSSV in shrimp with suspected infection (*OIE, 2009, 2019*). In terms of economic impacts, previous studies have estimated the economic loss from WSD to be more than 8 billion USD (*Stentiford, Bonami & Alday-Sanz, 2009*; *Verbruggen et al., 2016*). In Asia, the impacts from WSD were estimated to be 11 million USD in 2015 from production and economic losses (*Shinn et al., 2018*).

Possible risk factors have been reported in several studies, including: wastewater from a contaminated processing plant (*Reddy, Jeyasekaran & Shakila, 2013*), pond location near a sea (*Corsin et al., 2001*), feeding infected carriers, sharing of the water source with infected farms, high stocking density of shrimp (*Tendencia, Bosma & Verreth, 2011*), appearance of previously infected crops and seawater use (*Yaemkasem et al., 2017*), owners who own multiple farms (*Piamsomboon, Inchaisri & Wongtavatchai, 2015*), low atmospheric temperature and high daily atmospheric temperature (*Piamsomboon, Inchaisri & Wongtavatchai, 2016*), presence of weed on the farm site and inappropriate concentration of ammonia and oxygen (*Talukder et al., 2021*), and insecure post-larvae (PL) provider (*Worranut et al., 2018*). However, only a few studies have applied analytical techniques to understand factors that may vary over time. In this study, we used the longitudinal analysis of a targeted study area to improve understanding of the behavior of WSD.

Longitudinal studies can determine the association of risk factors over an extended period. This kind of study can help researchers reduce recall bias, a major error in well-known designs such as ordinary case–control studies (*Lindberg et al., 2017*; *Cronin et al., 2020*). Likewise, a longitudinal study can help identify all possible changes in factors that may occur over time (*Fitzmaurice, Laird & Ware, 2012*). Statistical testing with longitudinal studies must consider numerous factors. Previously, common statistical methods such as the analysis of variance (ANOVA), multivariate analysis of variance (MANOVA), mixed-effect regression model (MRM), and generalized equation estimation (GEE) were used for analysis. In prospective longitudinal data, the advantage is the ability to detect factors to exposures with regards to presence, timing, and chronicity. However, there are issues about follow-up and missing data (*Caruana et al., 2015*).

Using the GEE model in longitudinal data can reduce statistical problems such underestimation of variability and type II statistical errors (*Caruana et al., 2015*). *Tang et al. (2022)* used GEE to analyzed longitudinal changes in the lungs associated with asthma. They found that patients with persistent asthma had lower forced expiratory

indices than patients with asthma resolution. Moreover, children with persistent asthma developed poorer lung function growth. *Santanasto et al. (2022)* studied longitudinal changes in lean mass, fat mass, and grip strength of African Caribbean men. They found that the acceleration in muscle strength decrease lead the acceleration in lean mass decrease by 10–15 years. *Gianfrancesco et al. (2019)* examined the relationship between smoking and rheumatoid arthritis, controlling for time-varying covariates. They found evidence that smokers were associated with higher levels of disease activity in rheumatoid arthritis.

Rayong province, located on the east coast of the Gulf of Thailand, is an important shrimp farming area in Thailand; however, it suffers from continuously WSD occurrence. To limit the economics loss caused by WSD, a longitudinal design was used to study farm characteristics, farm practices, and biosecurity measures that influenced WSD occurrence over time using a structured interview schedule. The results of this study provide new insight into the presence of WSD, and offer a reference for authorities in their attempts to improve the disease control program of WSD in Thailand.

## MATERIALS AND METHODS

### Study framework and ethical statement

A 2-year longitudinal observational study was performed wherein data were collected using a structured interview schedule administered between October 2017 and September 2019. The study area was Rayong province, Thailand, located on the east coast of the Gulf of Thailand. All shrimp farms were located in two districts: most were in the Klang district, with the remainder in Mueang Rayong district. A total of 270 ponds from 22 marine shrimp farms were included in this study.

Informed verbal consent was obtained before commencing the data and shrimp sample collection. All procedures involving participants were performed following the principles of the Declaration of Helsinki. In addition, this study was approved by the Institutional Animal Care and Use Committee (IACUC) of the Department of Fisheries (DoF), Thailand. The approval number is U1-05341-2559.

### Structured interview schedule

The structured interview schedule contained both open- and close-ended questions asked to respondents in face-to-face interviews. The structured interview schedule was reviewed by aquaculture ($n = 1$) and epidemiology ($n = 1$) experts by proofreading. Before commencing the data collection, the structured interview schedule was tested for corrections with the help of 20 farmers. Later, the approved structured interview schedule was discussed with local fishery officers who performed the data collection. Each pond for the study was selected through a joint decision between the owner and the authors. However, the same pond or a different pond on the same farm was used for consecutive data collection. These ponds represent farms used as cultivation sites. Moreover, each farm was studied for 2–23 sequential pondings that might overlap with previous or subsequent cultivation (Supplement 1).

Each farmer was interviewed twice using the structured interview schedule. Certain questions were asked during the initial stage of cultivation regarding pond preparation and activities during the stocking period to avoid recall bias. The remaining questions were asked after harvesting a few days later. Furthermore, all respondents were farm owners or farm managers. In the structured interview schedule, questions included those on farm information, farm practices, farm biosecurity, and possible risk factors associated with WSD (Supplement 2). All questions were designed based on literature reviews (*Piamsomboon, Inchaisri & Wongtavatchai, 2015*; *Yaemkasem et al., 2017*). In detail, the questions adopted from *Piamsomboon, Inchaisri & Wongtavatchai (2015)* were those related to farm area, culture area, number of pondings, water reserve area, source of farming water, owner of multiple farms, adjacent farms, water management, and application of fertilizer. Questions derived from *Yaemkasem et al. (2017)* included those on biosecurity measures, farm characteristics, stocking density, feeding management, geographical factors, pond preparation, and water management. Finally, questions based on our own experiences in the study area included those on modernizing the water management system, the ponding system, factors related to weather and environment, source of PL, vector control, disease status, and laboratory confirmation of WSD. Additionally, the locations of the geographical centers of the farms were recorded and are illustrated in Fig. 1.

## Shrimp sampled status for WSD

To confirm the status of WSD, we collected PL and shrimp samples twice from each pond. First, PL was sampled before cultivating. In this study, only PL aged more than 10 days was collected. At least 400 PL in batches of 50 were collected and tested for WSD using nested polymerase chain reaction (PCR; *OIE, 2009*; *OIE, 2019*) at a local DoF laboratory or a certified private laboratory by DoF. In addition, only half tail of PL was used for testing. Second, the shrimp were sampled before harvesting. At least 60 shrimp in each pond were collected and were tested using nested PCR. For testing, pleopod, gill, and hemolymph of shrimp were collected for WSSV testing (*OIE, 2009*, *2019*) at a local DoF laboratory. The number of samples collected was according to the DoF requirement. A WSD-positive pond showed positive PCR results for WSSV in PL and/or shrimp samples. In contrast, WSSV in PL and shrimp samples were absent in ponds with negative PCR results.

For nested PCR, briefly, the pooled sampled of shrimp was homogenized in DNAzol® reagent (Invitorgen, Thermo Fisher Scientific, Waltham, MA, USA). The primers 146F1 (5′ ACTACTAACTTCAG CCTATCTAG 3′)/146R1 (5′ TAATGCGGGTGTAAT GTTCTTACGA 3′), and 146F1/146R1 plus 146F2 (5′ GTAACTGCCCCTTCCATCTCCA 3′)/146R2 (5′ TACGGCAGCTGCTGCACCTTGT 3′) were respectively used for one-step and two step PCR. The PCR products were visualized in 1.5% agarose gel electrophoresis containing SYBR® Safe DNA gel stain (Invitorgen, Thermo Fisher Scientific, Waltham, MA, USA). The expected amplicon sizes were respectively 1,447 and 941 bp, for first and nested step reactions (*OIE, 2009*; *OIE, 2019*).

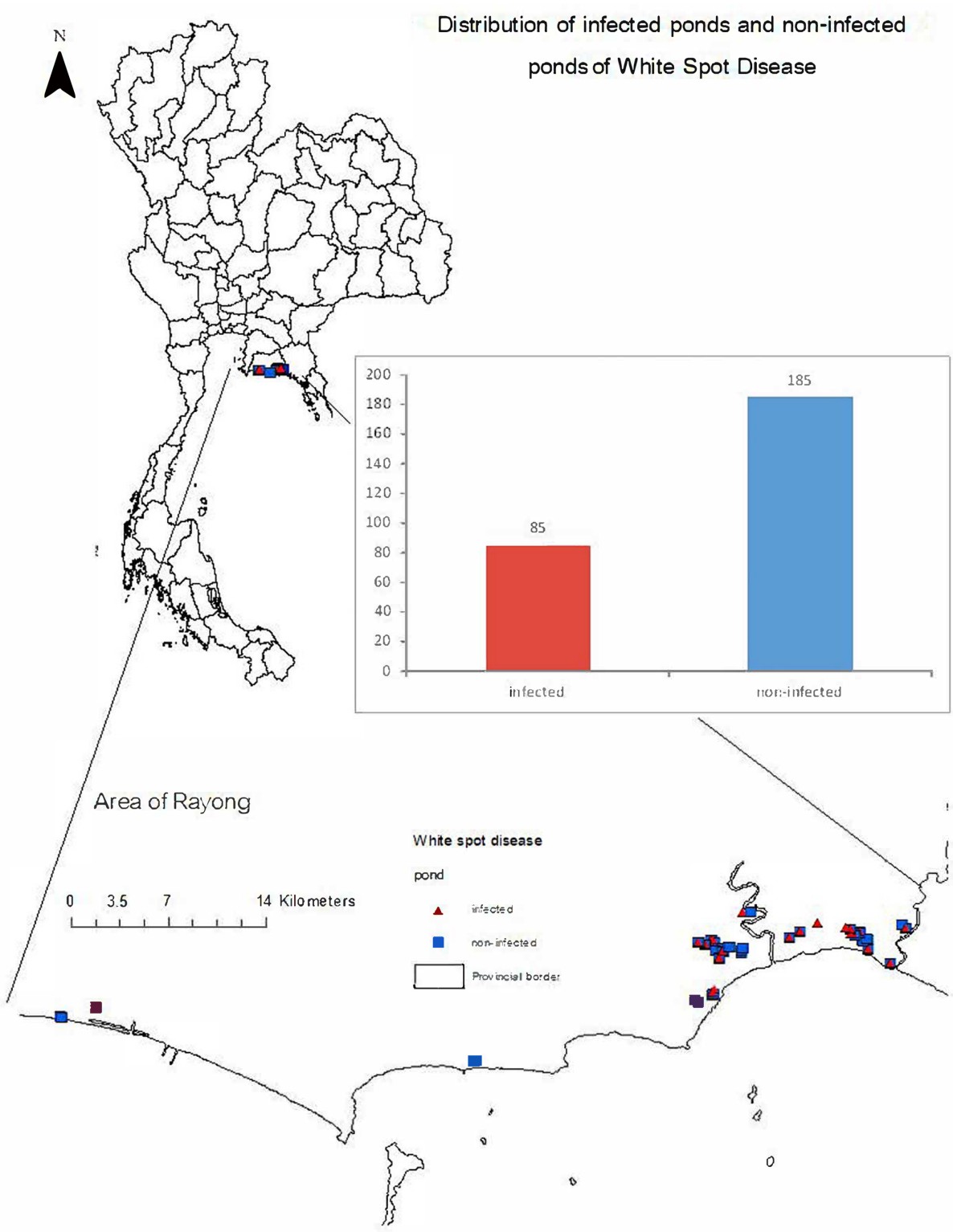

**Figure 1 Distribution of infected (symbol: red triangle) and non-infected (symbol: blue square) ponds associated with white spot disease (WSD) in Rayong, Thailand.**

## Data analysis

Univariate and multivariate analyses were performed using GEE with the R software version 4.1.1 (*R Core Team, 2021*) and geepack package (*Højsgaard, Halekoh & Yan, 2006*). This approach is a widely used statistical method in the analysis of longitudinal data. Here, the variance function was used as a binomial family to fit the logistic model. A covariance matrix was used as an exchangeable correlation structure. Through univariate analysis, possible risk factors were calculated in succession as independent variables. The WSD status was set as the dependent variable. Using multivariate analysis with backward elimination, the significant variables with an alpha level of 1% from univariate analysis were input to the GEE model to identify the association between possible risk factors and WSD. The final model reported significant factors with an alpha level of 1%. In this step, we performed several analyses and checked that each factor executed biological feasibility until a final model was obtained.

## RESULTS

### General information

From the 270 responses, we found that most farms raised shrimp by the monoculture of *Penaeus vannamei* (*P. Vannamei*). Most farms (217; 80.37%) had experience of over 10 years in marine shrimp farming. All farms were operated by multiple workers, including owners. In addition, 143 (52.96%) of owners have more than one farm. The number of crops per year ranged between one to five crops. There were 126 (46.67%) and 123 (45.56%) farms that had three and two crops per year, respectively. During the study, only four (1.48%) crops from two (9.09%) farms were raised in a monoculture of *P. monodon*. We found that the PL source varied from hatcheries within Rayong and neighboring provinces such as Trad, Chachoengsao, and Chonburi. On average, the price per PL was 0.16 Baht (0.09–0.50 Baht; 1 Thai Baht is approximately 0.03 US Dollar). Generally, most farmers raised PL at 12 days old (12–36 days). For WSD status, 85 (31.48%) of crops from 269 farms were infected (Fig. 1). Only one farm with nine crops did not have WSD during the study period. Example lesions of WSD in infected shrimp and positive results of WSSV are shown in Fig. 2.

### Univariate and multivariable analyses

After data management, 86 possible risk factors (categorical data = 71, continuous data = 15) were analyzed from univariate analysis using the GEE model. We found that 17 possible risk factors were statistically significant and associated with WSD status (Table 1). Subsequently, multivariate analysis was performed using the GEE model. We found that two independent variables were statistically significant with biological sense, corresponding to WSD status. These two categorical variables were: (1) inclement weather during ponding and (2) separation of the ponds into a cultured pond, water treatment pond, and water reservoir for ponding.

From Table 2, we found that the absence of inclement weather during ponding was a statistically significant factor associated with the occurrence of WSD. This factor prevents WSD occurrence as the odds ratio (OR) was lesser than one (OR = 0.196, 99% confidence

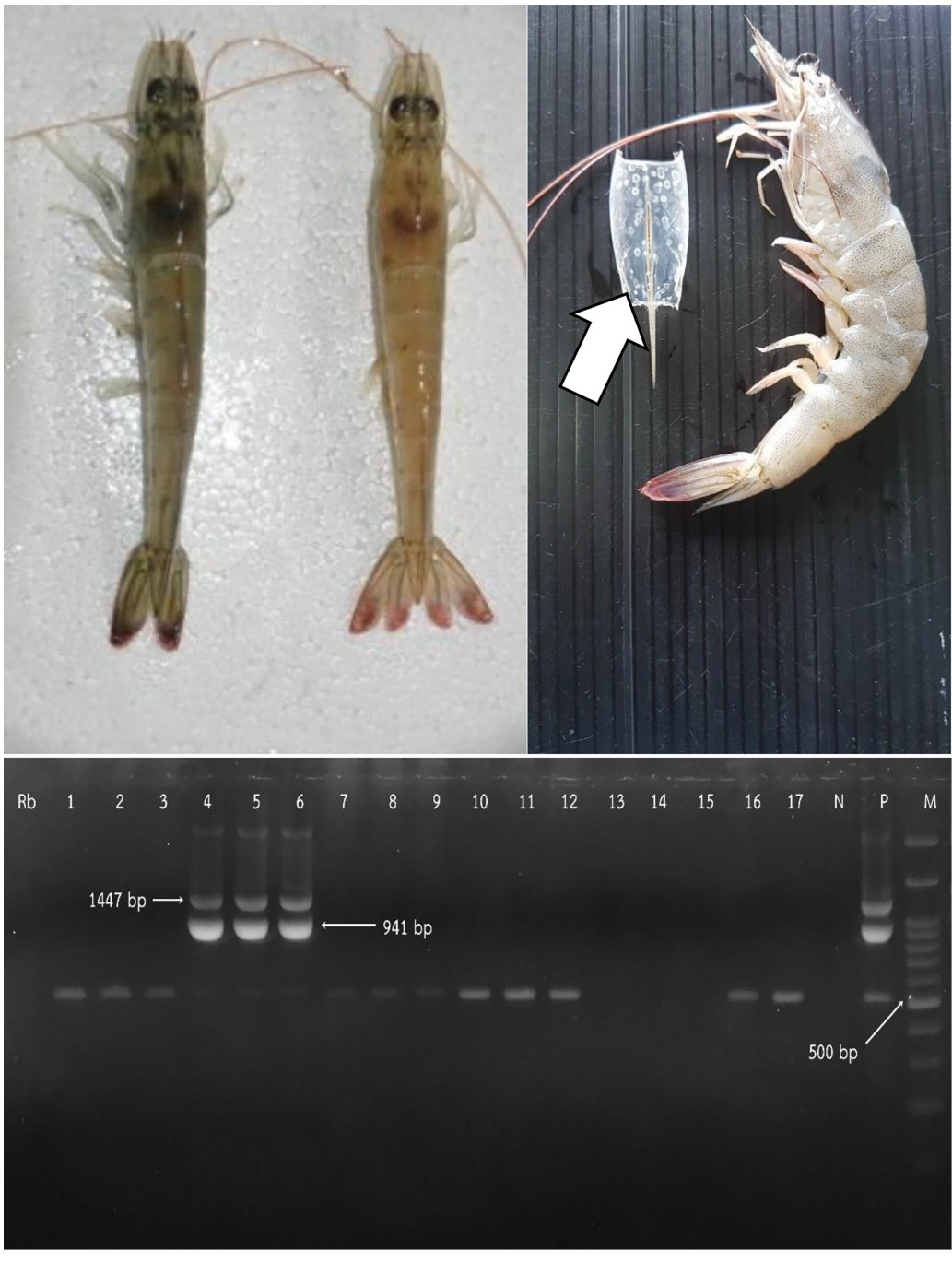

**Figure 2 Comparison of normal whiteleg shrimp and infected whiteleg shrimp discolored by white spot disease (WSD) (upper left). White spot lesions on carapace of shrimp (upper right). Agarose gel electrophoresis of positive white spot syndrome virus (WSSV) (below).**

interval [CI] of OR [0.041–0.938], $P = 1.3 \times 10^{-5}$). In contrast, inclement weather during ponding has been identified as a risk factor for WSD by approximately five times ($1/0.196 = 5.102$) compared with normal weather. Moreover, the separation of ponds into three specific types (cultured, water treatment, and pond for water reservoir) was

**Table 1 Univariate analysis of significant risk factors associated with white spot disease (WSD) in marine shrimp farms in Rayong, Thailand.**

| Possible risk factors | Type of variable (number of factor levels) | Mean (SD) | Number of exposures without WSD infection | Number of exposures with WSD infected | P-value |
|---|---|---|---|---|---|
| Farm size (unit: acres) | Numeric | 138 (252) | – | – | 0.00072 |
| A number of shrimp ponds | Numeric | 13 (16) | – | – | 0.0034 |
| Source of post-larvae (source Z) | Categorical (21) | | 11 | 3 | Reference |
| Source A | | | 0 | 1 | $<2 \times 10^{-16}$ |
| Source E | | | 2 | 0 | $<2 \times 10^{-16}$ |
| Source G | | | 3 | 0 | $<2 \times 10^{-16}$ |
| Source H | | | 0 | 2 | $<2 \times 10^{-16}$ |
| Source I | | | 1 | 0 | $<2 \times 10^{-16}$ |
| Source K | | | 0 | 2 | $<2 \times 10^{-16}$ |
| Source L | | | 2 | 0 | $<2 \times 10^{-16}$ |
| Source M | | | 2 | 0 | $<2 \times 10^{-16}$ |
| Source O | | | 0 | 8 | $<2 \times 10^{-16}$ |
| Source T | | | 2 | 0 | $<2 \times 10^{-16}$ |
| Size of post-larvae (larvae aged 12) | Categorical (6) | | 168 | 82 | Reference |
| Larvae aged 15 days | | | 9 | 0 | $<2 \times 10^{-16}$ |
| Larvae aged 22 days | | | 1 | 0 | $<2 \times 10^{-16}$ |
| Larvae aged 25 days | | | 1 | 2 | $<2 \times 10^{-16}$ |
| Larvae aged 30 days | | | 5 | 1 | $1.7 \times 10^{-5}$ |
| Larvae aged 36 days | | | 1 | 0 | $<2 \times 10^{-16}$ |
| Number of farms within 1 km radius from studied farm (one farm) | Categorical (5) | | 45 | 27 | Reference |
| Four farms | | | 13 | 1 | 0.0042 |
| Five farms | | | 0 | 1 | $<2 \times 10^{-16}$ |
| Size of sludge pond (unit: acres) | Numeric | 8.28 (16.7) | – | – | 0.0042 |
| Inclement weather during ponding (found) | Categories (2) | | 93 | 68 | Reference |
| Not found | | | 92 | 17 | $7.3 \times 10^{-5}$ |
| Usage of formalin during ponding (used) | Categories (2) | | 176 | 83 | Reference |
| Not used | | | 9 | 2 | $<2 \times 10^{-16}$ |
| Usage of benzalkonium chloride during ponding (used) | Categories (2) | | 180 | 85 | Reference |
| Not used | | | 5 | 0 | $<2 \times 10^{-16}$ |
| Usage of chemical fertilizers during ponding (not used) | Categories (2) | | 175 | 83 | Reference |
| Used | | | 10 | 2 | 0.0015 |
| Separation of water reservoir, usage of potassium permanganate, cultured pond and water treatment for ponding (not used) | Categories (2) | | 182 | 81 | Reference |
| Used | | | 3 | 4 | $5.6 \times 10^{-16}$ |
| Separation of cultured pond, water treatment pond and water reservoir for ponding (not used) | Categories (2) | | 160 | 83 | Reference |
| Used | | | 25 | 2 | 0.0075 |

| Possible risk factors | Type of variable (number of factor levels) | Mean (SD) | Number of exposures without WSD infection | Number of exposures with WSD infected | P-value |
|---|---|---|---|---|---|
| Added water during ponding (added) | Categories (2) | | 170 | 61 | Reference |
| Not added | | | 15 | 24 | 0.0079 |
| Cultivated water treatment before released into public resource (treated) | Categories (2) | | 175 | 60 | Reference |
| Not treated | | | 10 | 25 | 0.0078 |
| Usage of chlorine during ponding (not used) | Categories (2) | | 185 | 82 | Reference |
| Used | | | 0 | 3 | $<2 \times 10^{-16}$ |
| Usage of trichlorfon during ponding (not used) | Categories (2) | | 183 | 64 | Reference |
| Used | | | 2 | 21 | 0.00052 |
| Partial reusing of water for ponding (not reused) | Categories (2) | | 124 | 74 | Reference |
| Reused | | | 61 | 11 | 0.0014 |

**Note:**
P-value reported as $<2 \times 10^{-16}$ when the P-value produced by the software is too small to be significant.

**Table 2 Significant risk factors associated with white spot disease (WSD) in marine shrimp farms in Rayong, Thailand ($n = 270$), using multivariate analysis.**

| Factors (type of variable/number of factor levels) and significance factor level | Number of exposures without WSD infection | Number of exposures with WSD infected | Estimate | SE | OR | 99% CI of OR | P-value |
|---|---|---|---|---|---|---|---|
| Intercept | – | – | 0.0411 | 0.2778 | – | – | 0.882 |
| Inclement weather during ponding (categories/2) | | | | | | | |
| Found | 93 | 68 | – | – | – | – | Reference |
| Not found | 92 | 17 | −1.6286 | 0.373 | 0.196 | [0.041–0.938] | $1.3 \times 10^{-5}$ |
| Separation of cultured pond, water treatment pond and water reservoir for ponding (categories/2) | | | | | | | |
| No | 160 | 83 | – | – | – | – | Reference |
| Yes | 25 | 2 | −2.4916 | 0.7559 | 0.0828 | [0.0006–10.6688] | 0.001 |

**Note:**
SE = standard error, OR = odds ratio, 99% CI of OR = 99% confidence interval of odds ratio. Degrees of freedom is 267.

significantly associated with WSD occurrence. Likewise, the presence of this ponding system was a protective factor for WSD (OR = 0.0828, 99% CI of OR [0.0006–10.6688], $P = 0.001$). In contrast, other ponding systems have been identified as a risk factor for WSD by approximately 12 times ($1/0.0828 = 12.0773$) compared with pond separation into the three types described above.

## DISCUSSION

In this longitudinal study, we used GEE to analyze possible risk factors associated with WSD from 85 infected crops and 185 non-infected crops in Rayong, Thailand. Additionally, we found that the presence of inclement weather during ponding and the separation of ponds into a cultured pond, water treatment pond, and pond for water reservoir were highly significant risk factors corresponding to WSD occurrence.
In this study, most owners operated more than one farm and played an important role in each ponding. We found that the same owner with different farms had WSD-infected and non-infected ponding in different crops. In addition, most farms were infected and non-infected in the same pond at different times of ponding. This may indicate that WSD is not influenced by owner activities. The results of univariate analysis were in agreement with this finding, but are inconsistent with that of a previous study in Chanthaburi (a neighboring area of Rayong; *Piamsomboon, Inchaisri & Wongtavatchai, 2015*). In this study, most of the farm owners operated multiple farms. Of these, most managed the farms without infection of WSD. As such, our univariate analysis results are not statistically significant. In the Chanthaburi study, most owners operated a single farm. Theoretically, an area with a higher number of owners who operate multiple farms should be at greater risk of disease. It is possible that there are additional risk factors that were not measured in this study. Alternatively, we used a 1% chance of rejecting the null hypothesis in the GEE model because we wanted to identify factors with a strong association between independent and dependent variables; this may be why some factors (*e.g.*, owners operating multiple farms) were dropped from our analysis. Another possible reason is that the study area is major endemic area of WSD, and outbreaks spread rapidly. This could influence owners with multiple farms to change their behavior to reduce the risk of disease during an outbreak. Further study is needed to address each of these possibilities. Regardless, owners and other workers need to be careful to prevent contamination or infection between farming sites. Restrictions on biosecurity are a good way to control or limit WSD occurrence. Furthermore, our results show that only one farm did not have WSD occurrence during our study because this farm was operated with the highest biosecurity measures (complete closed-system farming). However, notably, this farm had expensive biosecurity operations, which are difficult to operate on general farms.

In the univariate analysis, we identified some significance factors with an alpha level of 1% that were inconsistent with the real world situation. For example, based on our repeated calculation of the GEE model, the source of PL (or the PL provider) had a negative value for some sources. This suggests that the PL provider should have a preventive impact on the occurrence of WSD, when in fact we actually identified a higher proportion of infections (data not shown). Other confounding factors included the PL size and number of farms in a 1 km radius of the studied farm. Therefore, we performed a backward elimination in which all non-biological sense factors were removed from the final model. This ensured that our final model had high accuracy.

The presence of inclement weather during ponding was a risk factor associated with WSD ($P = 1.3 \times 10^{-5}$), which is consistent with the study in Chanthaburi (*Piamsomboon, Inchaisri & Wongtavatchai, 2016*) and with those in other countries (*Hanson et al., 2008*; *Tendencia et al., 2010*). *Millard et al. (2021)* reported that abiotic factors (*e.g.*, extreme weather events) can increase the severity of WSD in shrimp ponds without careful management. Another study reported that climate change factors (*e.g.*, increasing temperature and acidification in marine systems) positively affects hosts' susceptibility to WSSV (*Shields, 2019*). *Sun et al. (2014)* reported that WSSV overworked the cellular

machinery used for its own propagation after the culture temperature rose from 18 °C to 25 °C. However, this factor needs further investigation. In our structured interview schedule we asked all respondents about extreme inclement weather during each ponding to prevent error and uncertainty when an owner could not decide when the weather was good or bad. In general, owners uses the principle of the presence or absence of monsoon during ponding. Many factors related to climate, such as atmospheric temperature, humidity, atmospheric pressure, wind speed, and amount of rainfall are measured by relevant authorities, but the study region covers an extensive area. Although the results of this study are sufficient to inform farmers and relevant authorities about the relationship between WSD occurrence and inclement weather, we highly recommend that further investigations be conducted on the weather at the pond level.

Some farmers raise shrimp by separating specific ponds during cultivation, resulting in a protective factor ($P = 0.001$). In this system, for each crop, the farm comprises an intensive grow-out pond, water treatment pond, and reservoir pond for adding water during cultivation, which is often used to reduce the risk of WSD and contamination through carriers. This is a closed system that has higher biosecurity than an open ponding system. According to a previous study, in terms of operational cost, farmers can invest and profit in this system (*Karim et al., 2014*). In other systems, most farmers have only cultured ponds and reservoirs. Water treatment is performed during pond preparation. Generally, farmers need to add water during cultivation; hence, they use water directly from natural reservoirs or reservoir ponds, increasing the risk of WSD occurrence through untreated water. This result is consistent with our previous study, indicating that water added without treatment during ponding is a major risk factor (*Yaemkasem et al., 2017*). In contrast, a closed recirculation system can reduce the risk of WSD more than an ordinary closed system; however, the operation of this system is more expensive. In our opinion, water management using a closed system is sufficient to limit the occurrence of WSD. However, the restrictions on biosecurity along with this system must be consistent.

In this study, the final model and univariate analysis using GEE showed a strong relationship with an alpha level of 1% between the dependent and independent variables, indicating that our analysis provides high-precision results. However, concerning the separation of ponds, we found a high variation of 99% confidence interval of odds ratio (OR), indicating that an uncertainty level is present around this factor. To prevent this possible error in the field, farmers and relevant authorities need to focus on the water added during cultivation. Every time the water is added, farmers must ensure the safety and efficiency of the treated water for WSSV. The results of this study can be applied to other areas in which WSD is present; however, shrimp farming in different geographic areas may have different geographical risk factors. We suggest that longitudinal data specific to a given area are needed to prevent potential losses from the outbreak of WSD.

## CONCLUSION

This study showed two statistically significant factors associated with WSD occurrence in Rayong, Thailand: inclement weather during ponding and separation of ponds during

cultivation. The findings of this study offer a reference for farmers and relevant authorities focused on possible risk factors during the shrimp cultivation period. In addition, the results could be advantageous in limiting WSD occurrence in Rayong, which is an endemic area for this disease. Finally, the results of this study could be applied to other areas in which WSD is present.

## ACKNOWLEDGEMENTS

We would like to thank all the officers of the Rayong Coastal Aquaculture Research and Development Center for their support.

### Funding

This study was financially supported by the Kasetsart Veterinary Development Fund of the Faculty of Veterinary Medicine, Kasetsart University, Thailand (code: 62.02) and Thailand's Department of Fisheries (Code: 63-3-0106-63049). The funders had no role in study design, data collection and analysis, decision to publish, or preparation of the manuscript.

### Grant Disclosures

The following grant information was disclosed by the authors:
Kasetsart Veterinary Development Fund of the Faculty of Veterinary Medicine, Kasetsart University, Thailand: 62.02.
Thailand's Department of Fisheries: 63-3-0106-63049.

### Competing Interests

The authors declare that they have no competing interests.

### Author Contributions

- Sompit Yaemkasem conceived and designed the experiments, performed the experiments, analyzed the data, prepared figures and/or tables, authored or reviewed drafts of the paper, and approved the final draft.
- Visanu Boonyawiwat conceived and designed the experiments, authored or reviewed drafts of the paper, and approved the final draft.
- Manakorn Sukmak conceived and designed the experiments, authored or reviewed drafts of the paper, and approved the final draft.
- Sukanya Thongratsakul analyzed the data, prepared figures and/or tables, authored or reviewed drafts of the paper, and approved the final draft.
- Chaithep Poolkhet conceived and designed the experiments, performed the experiments, analyzed the data, prepared figures and/or tables, authored or reviewed drafts of the paper, and approved the final draft.

## Animal Ethics

The following information was supplied relating to ethical approvals (*i.e.*, approving body and any reference numbers):

The Institutional Animal Care and Use Committee (IACUC) of the Department of Fisheries (DoF), Thailand. The approval number is U1-05341-2559.

## Data Availability

A questionnaire, code, and raw data are available in the Supplemental Files.

## Supplemental Information

Supplemental information for this article can be found online at http://dx.doi.org/10.7717/peerj.13182#supplemental-information.

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
