# Peer review of "A longitudinal study of risk factors associated with white spot disease occurrence in marine shrimp farming in Rayong, Thailand"

_PeerJ, doi:10.7717/peerj.13182_

## Round 0.1 · original submission · Major Revisions

Dear Dr. Yaemkasem and colleagues:

Thanks for submitting your manuscript to PeerJ. I have now received two independent reviews of your work, and as you will see, the reviewers raised some concerns about the research. Despite this, this the reviewers are optimistic about your work and the potential impact it will have on research studying white spot disease. Thus, I encourage you to revise your manuscript, accordingly, taking into account all of the concerns raised by both reviewers.

While the concerns of the reviewers are relatively minor, this is a major revision to ensure that the original reviewers have a chance to evaluate your responses to their concerns.

Importantly, please ensure your Materials and Methods are clearly stated. The methods should be clear, concise and repeatable. Please ensure this, and make sure all relevant information and references are provided. Please consider expanding the Conclusion (per reviewer 2’s suggestions).

There are many minor suggestions to improve the manuscript (typos, nuances, etc.).

I agree with many of the concerns of the reviewers, and thus feel that their suggestions should be adequately addressed before moving forward.

I look forward to seeing your revision, and thanks again for submitting your work to PeerJ.

Good luck with your revision,

-joe

Reviewer 1 ·

Basic reporting

Abstract: line 27, October 2007. Would it be 2017?

For introduction:
- I suggest that the authors should provide more crucial information about WSD situation such as severity, the impact to economy and the decision to study in Rayong province. The reader will better understand how important of this work.
- Since there were a few reports on the risk factor of WSD in Thailand carried out by different statistical analysis, so, I suggest that the authors should give the detail of pros and cons of your method compared to others.

Experimental design

Questionnaire part: the authors mention that you designed the questionnaire based on the research from Yaemkasem et al., 2017 and Piamsomboon, Inchaisriand & Wongtavatchai, 2015. I suggest that you should separately describe what the factors are from those previous works and what the new additional factors are in this study.

Shrimp sample collection and WSD status part: I suggest that the authors should briefly explain more about the organ for detection, DNA extraction step, PCR interpretation and the PCR results of the testing are from individual or pooled sample of an investigation, please specify.

Validity of the findings

- As the result of statistical analysis, the inclement weather was indicated as one of the risk factors of WSD. So, to figure it out, the authors should specify or give an example how the inclement weather be in those actual situations.
- Line 175-181: Once the authors discussed about the owner operation which is not influence to WSD in this study but inconsistent with the previous report. What is the different between previous study and yours? The study area? The statistical method? If it is because of the statistical method, which one is more accurate? Please discuss.
- The results need more figures about WSD status such as the shrimp showing WSD clinical signs and PCR positive result.
- In a nutshell, do you think that the risk factors obtained from Rayong model would also influence WSD in another shrimp cultured area in Thailand? Please discuss.

Additional comments

It is a good scientific study on the multiple risk factors of WSD occurrence. Also, it has a very well conducted with an approval of animal ethics. However, to make it more complete, some minor details need to be improved as I mentioned above.

Reviewer 2 ·

Basic reporting

The study reported is an interesting one and the longitudinal study has attempted to identify potential risk factors related to one of the most significant diseases, white spot disease caused by white spot syndrome virus (WSSV), affecting farmed shrimp. The study identified 17 statistically significant risk factors associated with WSSV. The information generated will be helpful in formulating management measures against WSSV. Following are the specific comments:

In the first part of the manuscript (Methods) as well as in Line No. 27 – The period of survey is mentioned as October 2007 – September 2019. However, this a two-year survey and it needs to be corrected as October 2017 - September 2019

Title: Can it be modified as A longitudinal study of risk factors associated with white spot disease occurrence in marine shrimp farming in Rayong, Thailand ?

Introduction: It has covered brief reviews. The statement of problem which highlights the gaps in past research studies has not been defined well. The researchers have mentioned that only few studies have applied analytical techniques to understand factors that may vary over time. It can be modified and rewritten with more empirical facts.

Materials and Methods:

Suitable research design has been applied in the present study.

The collection tool should be corrected as Interview Schedule instead of Questionnaire, as interview schedule is filled by researchers in face-to-face interaction whereas questionnaire is filled by the respondents.

The researchers have followed the standard method for developing Interview Schedule.

The data were collected only twice in two years. How many crop cycles have been covered under the study? A longitudinal study allows researchers to look at changes over time. Therefore, study must include the data on WSD status in consecutive crop cycles in the same farms.

Results:
The results are presented concisely.

If the 17 possible risk factors found in the study can be mentioned with their alpha level associated with WSD status, it would be useful to other researchers to conduct further research in the same line.

The study focussed only on farm characteristics, farm practices, and biosecurity measures in relation to the occurrence of white spot disease. However, do you think farm owners’/farm managers’ attributes such as knowledge, skill, experience, training status, innovativeness, etc. will have any bearing on the occurrence of white spot disease.

Discussion:
The results of the present study are well discussed in the light of past study findings.

Conclusion:
It is mentioned that “The findings of this study will help farmers and relevant authorities focus on possible risk factors during the shrimp cultivation period. In addition, it can be advantageous in limiting WSD occurrence in Rayong” . The authors may state whether these risk factors will be limited only to a particular location or it will be applicable to other shrimp-growing areas as well. The conclusion may be modified.

Experimental design

The study reported is an interesting one and the longitudinal study has attempted to identify potential risk factors related to one of the most significant diseases, white spot disease caused by white spot syndrome virus (WSSV), affecting farmed shrimp. The study identified 17 statistically significant risk factors associated with WSSV. The information generated will be helpful in formulating management measures against WSSV. Following are the specific comments:

In the first part of the manuscript (Methods) as well as in Line No. 27 – The period of survey is mentioned as October 2007 – September 2019. However, this a two-year survey and it needs to be corrected as October 2017 - September 2019

Title: Can it be modified as A longitudinal study of risk factors associated with white spot disease occurrence in marine shrimp farming in Rayong, Thailand ?

Introduction: It has covered brief reviews. The statement of problem which highlights the gaps in past research studies has not been defined well. The researchers have mentioned that only few studies have applied analytical techniques to understand factors that may vary over time. It can be modified and rewritten with more empirical facts.

Materials and Methods:

Suitable research design has been applied in the present study.

The collection tool should be corrected as Interview Schedule instead of Questionnaire, as interview schedule is filled by researchers in face-to-face interaction whereas questionnaire is filled by the respondents.

The researchers have followed the standard method for developing Interview Schedule.

The data were collected only twice in two years. How many crop cycles have been covered under the study? A longitudinal study allows researchers to look at changes over time. Therefore, study must include the data on WSD status in consecutive crop cycles in the same farms.

Results:
The results are presented concisely.

If the 17 possible risk factors found in the study can be mentioned with their alpha level associated with WSD status, it would be useful to other researchers to conduct further research in the same line.

The study focussed only on farm characteristics, farm practices, and biosecurity measures in relation to the occurrence of white spot disease. However, do you think farm owners’/farm managers’ attributes such as knowledge, skill, experience, training status, innovativeness, etc. will have any bearing on the occurrence of white spot disease.

Discussion:
The results of the present study are well discussed in the light of past study findings.

Conclusion:
It is mentioned that “The findings of this study will help farmers and relevant authorities focus on possible risk factors during the shrimp cultivation period. In addition, it can be advantageous in limiting WSD occurrence in Rayong” . The authors may state whether these risk factors will be limited only to a particular location or it will be applicable to other shrimp-growing areas as well. The conclusion may be modified.

Validity of the findings

The study reported is an interesting one and the longitudinal study has attempted to identify potential risk factors related to one of the most significant diseases, white spot disease caused by white spot syndrome virus (WSSV), affecting farmed shrimp. The study identified 17 statistically significant risk factors associated with WSSV. The information generated will be helpful in formulating management measures against WSSV. Following are the specific comments:

In the first part of the manuscript (Methods) as well as in Line No. 27 – The period of survey is mentioned as October 2007 – September 2019. However, this a two-year survey and it needs to be corrected as October 2017 - September 2019

Title: Can it be modified as A longitudinal study of risk factors associated with white spot disease occurrence in marine shrimp farming in Rayong, Thailand ?

Introduction: It has covered brief reviews. The statement of problem which highlights the gaps in past research studies has not been defined well. The researchers have mentioned that only few studies have applied analytical techniques to understand factors that may vary over time. It can be modified and rewritten with more empirical facts.

Materials and Methods:

Suitable research design has been applied in the present study.

The collection tool should be corrected as Interview Schedule instead of Questionnaire, as interview schedule is filled by researchers in face-to-face interaction whereas questionnaire is filled by the respondents.

The researchers have followed the standard method for developing Interview Schedule.

The data were collected only twice in two years. How many crop cycles have been covered under the study? A longitudinal study allows researchers to look at changes over time. Therefore, study must include the data on WSD status in consecutive crop cycles in the same farms.

Results:
The results are presented concisely.

If the 17 possible risk factors found in the study can be mentioned with their alpha level associated with WSD status, it would be useful to other researchers to conduct further research in the same line.

The study focussed only on farm characteristics, farm practices, and biosecurity measures in relation to the occurrence of white spot disease. However, do you think farm owners’/farm managers’ attributes such as knowledge, skill, experience, training status, innovativeness, etc. will have any bearing on the occurrence of white spot disease.

Discussion:
The results of the present study are well discussed in the light of past study findings.

Conclusion:
It is mentioned that “The findings of this study will help farmers and relevant authorities focus on possible risk factors during the shrimp cultivation period. In addition, it can be advantageous in limiting WSD occurrence in Rayong” . The authors may state whether these risk factors will be limited only to a particular location or it will be applicable to other shrimp-growing areas as well. The conclusion may be modified.

---

## Round 0.2 · accepted · Accept

Dear Dr. Yaemkasem and colleagues:

Thanks for revising your manuscript based on the concerns raised by the reviewers. I now believe that your manuscript is suitable for publication. Congratulations! I look forward to seeing this work in print, and I anticipate it being an important resource for groups studying white spot disease. Thanks again for choosing PeerJ to publish such important work.

Best,

-joe

Reviewer 1 ·

Basic reporting

no comment

Experimental design

no comment

Validity of the findings

no comment

Additional comments

I appreciated that the author and team have revised manuscript from all my comment very well. I have no doubt on any parts of the manuscript.

Reviewer 2 ·

Basic reporting

The manuscript is presented in an acceptable standard.

Experimental design

Research question well defined and relevant.

Validity of the findings

Findings are valid and has practical relevance.

Additional comments

The authors have addressed all the comments made by the reviewer and the manuscript has been improved. Therefore, the manuscript is recommended for acceptance.